# Sensor Reliability in Cyber-Physical Systems Using Internet-of-Things Data: A Review and Case Study

**Fernando Castaño** [1],*, **Stanisław Strzelczak** [2], **Alberto Villalonga** [1], **Rodolfo E. Haber** [1] and **Joanna Kossakowska** [2]

[1]  Centre for Automation and Robotics (CSIC–UPM), Spanish National Research Council, Arganda del Rey, 28500 Madrid, Spain; alberto.villalongaj@alumnos.upm.es (A.V.); rodolfo.haber@car.upm-csic.es (R.E.H.)

[2]  Faculty of Production Engineering, Warsaw University of Technology, 02-524 Warsaw, Poland; s.strzelczak@wip.pw.edu.pl (S.S.); j.kossakowska@pw.edu.pl (J.K.)

*  Correspondence: fernando.castano@car.upm-csic.es; Tel.: +34-918-70-50

**Abstract:** Nowadays, reliability of sensors is one of the most important challenges for widespread application of Internet-of-things data in key emerging fields such as the automotive and manufacturing sectors. This paper presents a brief review of the main research and innovation actions at the European level, as well as some on-going research related to sensor reliability in cyber-physical systems (CPS). The research reported in this paper is also focused on the design of a procedure for evaluating the reliability of Internet-of-Things sensors in a cyber-physical system. The results of a case study of sensor reliability assessment in an autonomous driving scenario for the automotive sector are also shown. A co-simulation framework is designed in order to enable real-time interaction between virtual and real sensors. The case study consists of an IoT LiDAR-based collaborative map in order to assess the CPS-based co-simulation framework. Specifically, the sensor chosen is the Ibeo Lux 4-layer LiDAR sensor with IoT added capabilities. The modeling library for predicting error with machine learning methods is implemented at a local level, and a self-learning-procedure for decision-making based on Q-learning runs at a global level. The study supporting the experimental evaluation of the co-simulation framework is presented using simulated and real data. The results demonstrate the effectiveness of the proposed method for increasing sensor reliability in cyber-physical systems using Internet-of-Things data.

**Keywords:** Cyber-Physical Systems; reliability assessment; Internet-of-Things; LiDAR sensor; driving assistance; obstacle recognition; reinforcement learning; Artificial Intelligence-based modelling

## 1. Introduction

Nowadays, the precise knowledge of the most appropriate sensor operating conditions and fault detection systems are among the cornerstones of scientific and technical studies for automated systems [1]. These are based upon on-line monitoring processes and additional comprehensive interpretation of sensor data by assessing sensor reliability. Sensors are driving the rapid growth of Cyber-Physical Systems (CPSs) and the Internet of Things (IoT) [2]. Both paradigms are pushing towards the next generation of sensor networks and unpredictable future applications, meaning that sensor reliability has become one of the most important and desirable performance indicators in the design, implementation, and deployment of future sensor networks [3,4].

An important reliability-related issue to be detected in autonomous systems, in order to self-correct problems such as lost data packages, and data collision, among others, is the failure of one network element [5–7]. One possible solution is to build real-time prediction models that maximize robustness and lifetime [8]. There are, in fact, several methods for the evaluation of sensor reliability. Each issue of

reliability, or that might affect it, can be assessed individually and as a whole, through a total error band figure. There are important features to be considered, such as sensitivity, range, precision, resolution, accuracy, offset, linearity, dynamic linearity, hysteresis and response time [9,10]. Evaluating sensor reliability includes probabilistic and statistical data that increase estimation reliability [11]. Evidence theory can be used, such as the Dempster-Shafer theory of belief functions. Quantifying reliability implies predictions concerning sensor lifetime and failure probability. Reliability can therefore be based on both statistical and Artificial Intelligence (AI) models. Suitable probability functions must be defined, which will be used to calculate the future behavior of devices, based either on carefully controlled laboratory experiments or on thorough failure analysis while in use. A typical product will be liable to various failure modes that change over time in a characteristic manner, so that the probability functions are themselves time dependent.

Currently, two of the most widely applied sensors are the 3D stereoscopic camera and the LiDAR [12,13]. Dozens of new applications have recently been reported [14–16]. However, the computational load and the processing time for data fusion of 3D stereoscopic camera and the LiDAR sensors in automotive applications are very high [17]. Therefore, a possible solution is to use only the point cloud provided by the LiDAR with a model for predicting the reliability of the sensor data and only activate the support sensors when a possible failure is anticipated [18].

Indeed, it is very difficult to guarantee the real topology and distance of objects with less uncertainty due to dead zones, object transparency, light reflection, weather conditions, and sensor failures, among others [19]. It is mandatory to obtain data in parallel from other connected LiDAR sensors to guarantee reliability. Therefore, new IoT-based technologies open up a wide range of methodologies for estimating and ensuring reliability in key emerging applications [20].

The most widely used techniques for modelling predictions in product lifetime and failure probability are probabilistic methods. Probabilistic methods for uncertain reasoning represent another group of techniques. Probability theory predicts events from a state of partial knowledge, while possibilistic models, represented by fuzzy systems, are applied to situations with intrinsic vagueness and uncertainty.

However, the prediction techniques are hardly limited to those mentioned above. Several clustering techniques such as nearest neighbor methods have been explored, in order to enable self-detection and self-correction capabilities [21]. Other capabilities to be considered from the perspective of reliability are self-adaptation and self-organization by embedding artificial neural networks (ANNs) in CPSs [22]. Efficient performance of multiple sensors and their online monitoring and self-correction procedures, through the application of machine learning (ML), such as Support Vector Machines (SVM) and ANNs, are essential for reducing maintenance costs, risk associated with uncalibrated and faulty sensors, increasing sensor reliability and, consequently, extended equipment life [23,24]. With the aim of guaranteeing certain safety and security conditions in some critical applications, the verification of sensory data and subsequent data evaluation are described in this paper on the basis of simulation of virtual and real scenarios, as well as a framework that properly combines both scenarios.

A reliability assessment procedure is therefore described in this paper that is applicable to data captured by IoT LiDAR sensors in automotive applications: LiDAR self-testing methodology. The reliability analysis is based on the paradigm of cyber-physical systems (CPS) by distributing nodes locally and globally, as will be explained later on. Each computing node has data-processing methods and machine-learning models for reliability prediction. In addition, a run-time self-learning and decision-making model runs within a global node, in order to determine the best model and the model updating mechanism on request.

This paper will be organized into five sections. Following this introduction, the second section will present a review of the state-of-the-art of the CPS-based reliability concept for sensor system reliability using AI methods. Subsequently, the specifications and the requirements obtained from the review of CPS reliability frameworks will be summarized in Section 3. A particular implementation of a CPS-based co-simulation framework will also be proposed in this section. In addition, a case study

for the evaluation of an IoT sensor network using a CPS-based co-simulation framework approach will be described in Section 4. In that section, the experimental results and a discussion relating to a comparative study will also be addressed. Finally, the conclusions and future research steps will be presented in Section 5.

## 2. Reliability of Sensors in Cyber-Physicals Systems. A Brief State-of-the-Art Review

Reliability of sensors has become not only the main focus of on-going worldwide research, but also the priority of research and innovation actions (RIA) supported by Horizon 2020 Programme. For the sake of giving only a brief overview, Table 1 summarizes some of the most challenging projects targeting sensor reliability, funded mainly by Electronic Components and Systems for European Leadership (ECSEL) Joint Undertaking [25] in the last five years. It is clearly shown that the reliability of sensors and remote sensing systems is a key enabling step towards massive utilization of sensor networks in all application fields from manufacturing up to maritime and aeronautic applications.

**Table 1.** Research and innovation actions in sensor reliability supported by H2020 Programme in the last five years.

| Horizon 2020 Initiative | Main Sensors | Application Field | Reference |
|---|---|---|---|
| RobustSENSE: Reliable, Secure, Trustable Sensors. For automated driving | Laser scanners, position, Virtual sensors | Automotive | Mäyrä et al. [26] |
| IoSense: Flexible FE/BE Sensor Pilot Line for the Internet of Everything | Microphones, force, pressure, gas, camera, LiDAR, accelerometer, others | Manufacturing, automotive, energy, environment | Castaño et al. [27] Godoy et al. [28] |
| SECREDAS: Cyber Security for Cross Domain Reliable Dependable Automated Systems | Cameras, position, ultrasounds, LiDAR, pressure | Automotive | Le et al. [29] |
| I-MECH: Intelligent Motion Control Platform for Smart Mechatronic Systems | Position, vision, force | Manufacturing, pharmaceutic, health | Valencia et al. [30] |
| PRYSTINE: Programmable Systems for Intelligence in Automobiles | Cameras, LiDAR, position, ultrasounds | Automotive | Druml et al. [31] Godoy et al. [32] |
| Power2Power: Providing next-generation silicon-based power solutions in transport and machinery for significant decarbonisation in the next decade | Temperature, PZT, radar, current, voltage, accelerometers | Manufacturing, energy, industrial machinery | Guerra et al. [33] La Fe et al. [34] |
| NeXOS: Next Generation Web-Enabled Sensors for the Monitoring of a Changing Ocean | optics, passive acoustics sensors, detectors | Environment | Toma et al. [35] |
| ReMAP: Integrated Fleet Management solution aimed at replacing fixed-interval inspections with adaptive condition-based interventions | Piezo-electric, acoustic emission, optical-fibber | Aeronautic | Lizé et al. [36] |

About twenty thousand scientific reports have considered reliability as the cornerstone of the main works [6,37]. The assessment of the sensor's reliability in a classification problem based on the transferable belief model was presented in [38], whereas the Dempster-Shafer theory of belief functions was the foundation for a framework in another seminal paper [11]. The reliability of leap motion sensor in static and dynamic tracking was analyzed in [39], with not-very-promising expectations due to limited sensory space and inconsistent sampling frequency. Special attention has been received by wireless sensor networks, considering a measure of reliability using a probabilistic graph [40], packet delivery mechanism for guaranteeing quality of service [41] and adaptive and cross-layer framework for reliable and energy-efficient data collection [42]. Some methods for improving reliability in data transmission and localization by combining particle and finite impulse response filtering [43] and a new routing protocol [44] have recently been reported. The reader may find an interesting review of reliability assessment of wireless sensor network for industrial applications in [45].

Cyber-physical systems pose more scientific and technical challenges due to the need for conjunctive approaches for evaluating sensor reliability using Internet-of-Things data [46,47]. A pioneering work suggested a method called Loss Inference based on Passive Measurement (LIPM) to

compute the link loss performance in digital ecosystems with plenty of distributed data sensors [48]. Scheduling policies formulated as a risk-sensitive Markov Decision Process were considered in an asymptotic approach to reliability [49]. Likewise, the Markov model for the reliability analysis of sensor nodes in wireless sensor networks was proposed in [3]. Similarly, the use of the stochastic optimization problem to maximize estimation reliability in water distribution networks was suggested in [50]. The quantification of the reliability of grid splitting under degraded communication conditions in the presence of cyber interdependencies was presented in [51]. Recently, a method based on the finite element model of a bridge integrated with structural vibration data for a self-governing structural health monitoring system was shown in [52]. A set of algorithms for estimating the reliability of the system under certain responsible time constraints was also recently suggested in [53]. Based on this review, it is evident that probabilistic-based approaches have been explored for addressing sensor reliability.

Certainly, the evaluation of sensor reliability is a highly important task, especially for Internet of Things (IoT) applications, and new performance indices need to be proposed [54]. Some recommendations for the standardized evaluation of sensor reliability are given in [55]. A recent review of analyses of the state of the art time delays, network size, energy efficiency, scalability, and reliability of mobile wireless sensor networks can be found in [56]. Attention has also been focused on lowering latency and improving protocols [57,58].

How can machine learning methods enable improvements in sensor reliability in cyber-physical systems?

The answer to this question is not easy, because the conditioning and re-elaboration of machine learning methods for assessing sensor reliability is not straightforward. This is indeed a new research trend with some pioneering reports in the early of 1990s [59]. Bayesian approaches [60,61], Fuzzy set theory [62,63], Dempster-Shafer (D-R) evidence theory [64–66], and Grey Group Decision-Making [67], among other methods, have recently been explored to address the reliability of sensors using Artificial Intelligence. However, many issues related with sensor reliability in cyber-physical systems using Internet-of-Things data remain unsolved, such as efficient conditioning and pre-filtering of big sensor data, high computational cost and limited parametrization of machine learning methods.

## 3. CPS-Based Reliability Approach

The truly challenging aspects of sensor network reliability and its evaluation have yet to prompt an exhaustive exploration and evaluation of sensory data under critical conditions. A gap is addressed in this study by sensors and sensor data in a CPS.

### 3.1. Sensor Reliability Assesment

One approach for assessing sensor reliability in key emerging applications, such self-driving in the automotive sector [5], is a model-based design for representing key sensor parameters. These parameters can be derived from monitoring processes where 'functional parameters' refer to the sensor characteristics and sensor lifetime, as well as to economic aspects (see Figure 1).

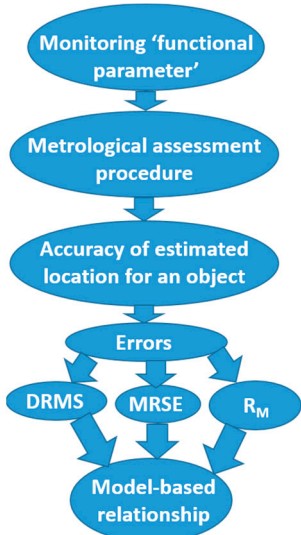

**Figure 1.** Procedure for sensor reliability assessment using a model-based approach for sensor data and key performance indices.

With the aim of increasing the reliability of data collected by LiDAR, metrological assessment procedures should also be applied. Linear interpolation of measurements from three detectors arranged in series is a time-saving procedure for processing and reducing LiDAR data [68].

All the major sources of potential error that influence point positioning accuracy have to be considered in the analytical derivations in order to determine the reliability of the achievable point positioning accuracy of LiDAR systems. Some of the random errors are reported in [69]. Some mathematical foundations are also provided for point positioning accuracy derived from the LiDAR equation, via error propagation:

$$r_M = r_{M,INS} + R_{INS}^M\left(R_L^{INS}{\cdot}r_L + b_{INS}\right) \tag{1}$$

where $r_M$ represents the 3D coordinates of an object point in the mapping frame; $r_{M,INS}$ represents time-dependent 3D INS coordinates in the mapping frame, provided by GPS/INS; $R_{INS}^M$ is the time-dependent rotation matrix between the INS body and the mapping frame; $R_L^{INS}$ is the boresight matrix between the laser frame and the INS body frame; $r_L$ represents the 3D object coordinates in the laser frame; and, $b_{INS}$ is the boresight offset vector.

In addition, the calculation of the accuracy of the estimated location of an object using the LiDAR sensor can be performed by other key performance indices. For example, the use of the Distance Root Mean Squared (DRMS) measure for the data that are tracked on the x-y plane (2D) and the Mean Radial Spherical Error (MRSE) measure for the data that are tracked in the x–y–z space (3D) were reported in [27,70]. Using derivable error formulas, any given random error and scan angle in the LiDAR range can be modelled and simulated. By doing so, the factors affecting LiDAR system accuracy can be analyzed [71].

### 3.2. Statistical and Artificial Intelligence-Based Methods

The Bayesian and Hidden Markov techniques are the most widely applied modeling techniques for reliability assessment under fuzzy environments [72,73]. A Bayesian network is a directed acyclic graph consisting of a set of nodes, representing random variables and a set of directed edges, representing their conditional dependencies. The dependencies in a Bayesian network can be adaptively determined from a dataset through a learning process. The objective of this training is to provide the best description of the probability distribution over the dataset because the attribute values are not supplied in the dataset [74].

In addition to those probabilistic methods, new tools have been reported in the literature, highlighting the use of Artificial Intelligence (AI) techniques and, in particular, Machine Learning (ML), to solve complex situations [34]. AI techniques also provide cognitive abilities, so that performance may be improved by increasing network lifetime and reliability [75]. Those techniques include ANN and fuzzy inference systems [76,77]. Zhang et al. proposed a soft-computing system based on Genetic Algorithm-Support Vector Regression (GA-SVR), in order to facilitate the reliability and survivability of the Structural Health Monitoring (SHM) system faced with, for example, an invalid fiber link in the sensor network [78].

## 4. CPS-Based Co-Simulation Framework

Some factors that can affect CPS reliability include component failure, environmental effects, task changes, and network update. A strategy for testing the reliability of CPSs and for their evaluation is proposed in [79] by analyzing both the internal and the external factors that influence their reliability. One solution could be to evaluate each element that constitutes the system: testing hardware, software, and architecture, as well as performance reliability including service reliability, cyber security reliability, resilience and elasticity reliability, and vulnerability reliability.

Behavioral simulations of CPS and IoT are increasing in their relevance with respect to analyzing reliability, because precise mathematical modeling is not straightforward [80]. These simulations are based on addressing four main topics: node localization, energy management, network multi-objective optimization, and self-capabilities approach [81,82].

While the reliability evaluation of physical systems is well understood and has been extensively studied, the reliability evaluation of a CPS is still under-studied, because CPS will not degrade the performance in the same way of physical systems and cannot be represented by a well-defined failure model. An evaluation framework is therefore required in order to assess the performance of CPS. A framework for CPS reliability analysis that includes reliability-based runtime reconfiguration is proposed in [83]. This framework is codified in a domain-specific modelling language that provides details on operational constraints and dependences.

However, domain-specific modelling-based analysis is, in many cases, unable to compute reliability functions efficiently (e.g., in terms of failure distributions) for complex systems. To do so, a frequency-domain reliability analysis framework of transportation CPSs was described in [84]. The advantage of that method is its capability to capture higher-order moments of the system characteristics, its scalability for the analysis of the reliability of complex systems, and efficient calculations.

In addition, it is important to consider the evaluation of other aspects of the CPS, such as safety and, particularly, security; different aspects still in the focus of research in the past few years. Therefore, the design of the simulation framework for CPS should consider these aspects at three levels: security objectives, security approaches and security in specific applications [85]. However, the cyber part not only should be secured, but also the physical part should take into account possible threats. A multi-cyber (computational unit) framework was compared with traditional models to improve the availability of the CPS based on the Markov model is reported in literature. The evaluation was carried out in terms of availability, downtime, downtime costs, and the reliability of the CPS framework [86]. Finally, another work considered an Internet-based computing platform in the form of a global computing node. In [87], a new cloud security management framework was introduced, based on improving collaboration between cloud providers, service providers, and service consumers for the management of cloud platform security and the hosting services. In addition, although in some applications this will not be possible, it is important to consider the possibility of introducing the human factor in the reliability analysis procedure. A human-interactive Hardware-In-the-Loop Simulation (HILS) framework for CPS was developed in [88] to support reliability and reusability in a fully distributed operating environment.

## *4.1. CPS-based Co-simulation Framework*

Based on the above contributions, and considering the initial list of requirements for the deploying an IoT sensor network, a CPS-based co-simulation framework is proposed. The IoT sensor network will supply physical data having (local and global) computing nodes for processing the sensory data.

### 4.1.1. Conceptual Scheme

In addition, the IoT sensor network has a global or main node composed of a knowledge database, a Q-learning method for decision-making, and an AI-based model library. During the simulation and the real running, a decision-making module will decide which specific model is the best in the current instant, taking into account the data received by all nodes that make up the network.

The functionalities are distributed in different nodes, both virtual and real, according to their functions. The distributed virtual or real nodes manage the capture of sensor data and run the error prediction calculation with the required accuracy, while the global or main node incorporates the runtime model that is generated, the library, and the knowledge database (see Figure 2).

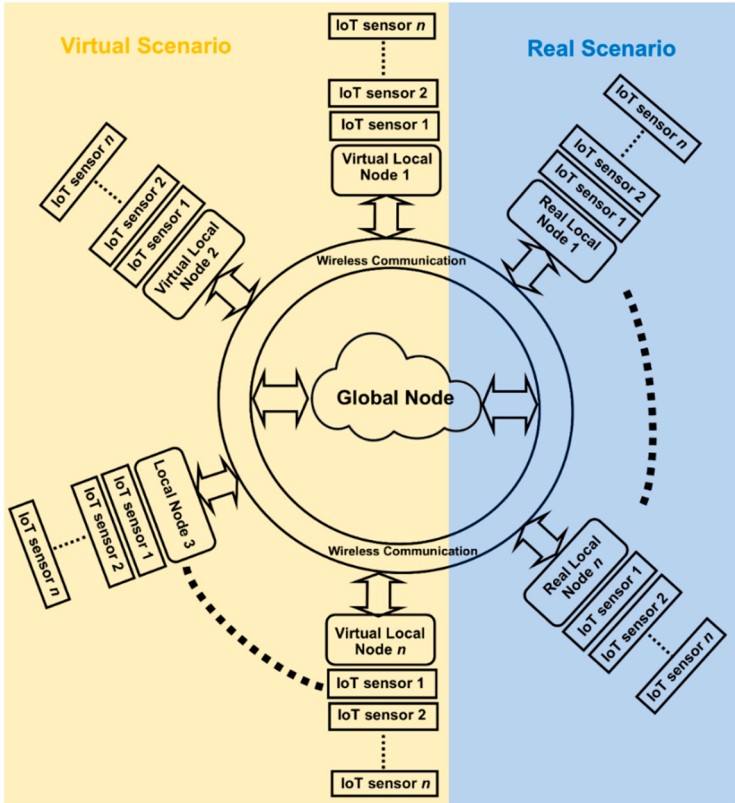

**Figure 2.** General scheme of the CPS-based co-simulation framework with virtual and real computing nodes and IoT sensor network.

The IoT sensors should be able to establish reliable and accurate wireless communications, ensuring that all the intrinsic challenges in an IoT network and in the different CPSs can be solved. This is achieved through the implementation of the architecture that is represented in Figure 2 by a network of *n* nodes, each node having *n* IoT sensors. In addition, the computing nodes communicate with each other and with their corresponding global node.

### 4.1.2. Procedure Description

The framework is designed taking into account that both the real and the virtual (local) computing nodes operate in parallel with the global computing node [89]. Data exchange between the different

nodes takes place in two different ways. On the one hand, data exchange between local nodes is produced in both the virtual (3D model simulation tool) and the real scenario. On the other hand, there is the data exchange between different local nodes and the global node using the 802.11p wireless communication protocol.

Therefore, there is an interaction between the simulated and the real environments and external applications that are running in the main node. Figure 3 shows the schematic diagram of the data exchange or messages exchange within the co-simulation framework.

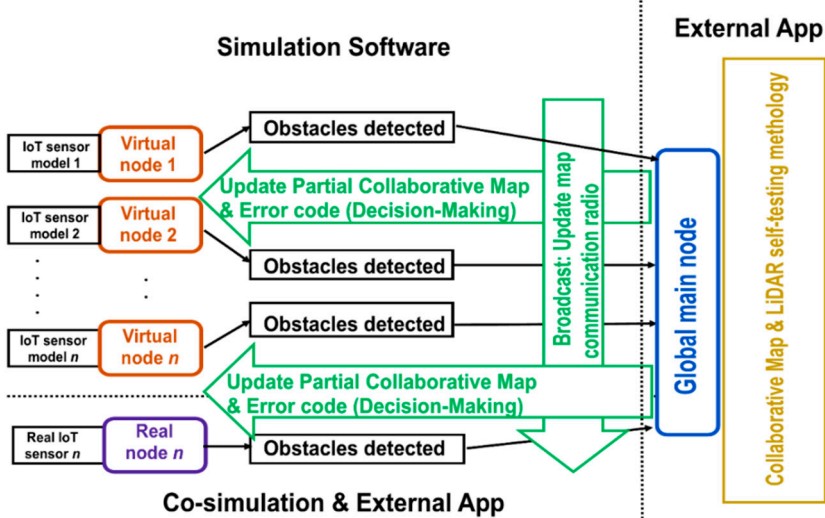

**Figure 3.** Conceptual flowchart showing the operation of a reliability co-simulation framework with CPS computing nodes and IoT sensors.

In this specific implementation, which is described more accurately in the following section, there is a wireless exchange of messages between different nodes using the 802.11p communication protocol in the following way. First, the local nodes with their IoT sensors detect different objects and their respective properties. Secondly, this information is shared on the network through a broadcast process.

## 5. IoT LiDAR-based Collaborative Mapping – A Case Study

The IoT sensor network chosen to evaluate the CPS-based co-simulation framework is composed of virtual and real LiDAR sensors [28]. An Ibeo Lux LiDAR 4-layer sensor was used with the following specifications: horizontal field of 120 deg, horizontal step of 0.125 deg., vertical field of 3.2 deg., vertical step of 0.8 deg., range of 200 m, and an update frequency of 12.5 Hz. As previously mentioned, the sensor network to be evaluated is composed of IoT sensors. The sensor network therefore has IoT capabilities connected to its computing nodes. These nodes are on-board computers integrated in an autonomous vehicle with a wireless communication interface between them.

The particular implementation of the CPS-based co-simulation framework, the LiDAR-based collaborative map is based on a co-simulation framework between two different software systems, designed in [90]. However, the contribution of this study is to include the real part in the co-simulation framework. This framework consists mainly of a computer-aided system to enable efficient interaction between the virtual scenario with virtual nodes setting in the simulation tool of Webots for automobile 8.6 [91] and an external application development for the computing nodes in the real scenario. The scenario in this particular case, in which the vehicles are represented as nodes, is as follows. A real vehicle (in a real scenario) and three virtual vehicles in the simulated scenario are detecting obstacles. Both kinds of vehicles share the position, object type and size of the obstacles (e.g., pedestrian, trees on the road and another vehicle). This is possible thanks to the IoT LiDAR network using an IoT obstacle

detection application (see Section 4.1), created in run time. Figure 4 shows the detailed diagram of the implementation of the LiDAR-based collaborative map using the CPS-based framework approach.

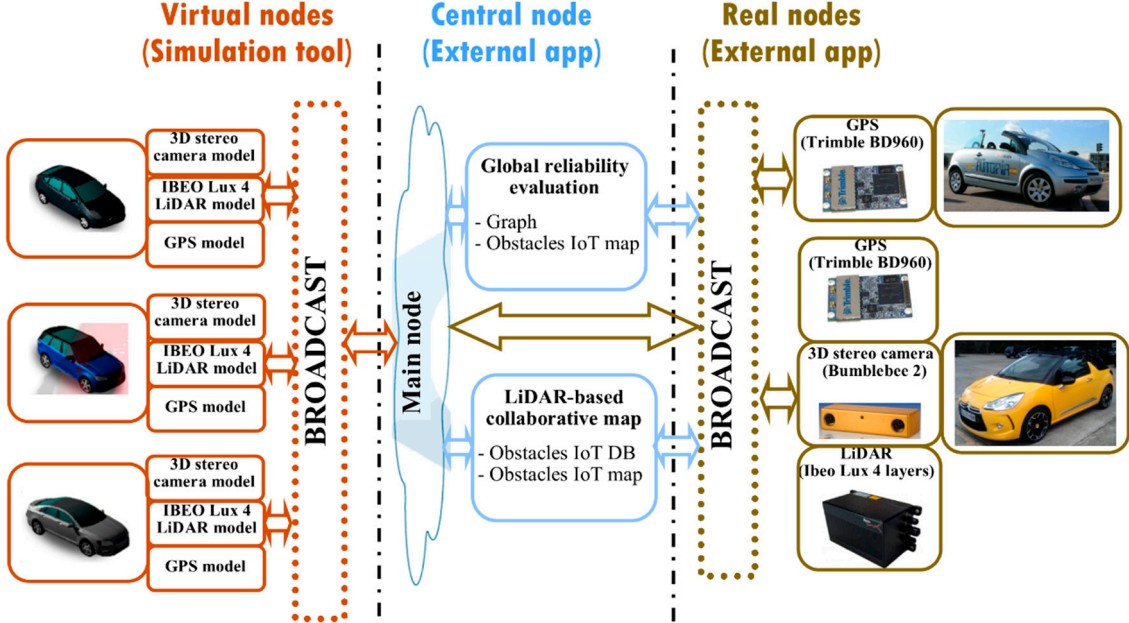

**Figure 4.** Detailed diagram of the implementation of the LiDAR-based collaborative map approached through a CPS-based co-simulation framework.

As mentioned above, the exchange of information packets between the local nodes with the main or global node is possible thanks to the use of a communication protocol using a UDP (User Datagram Protocol) as the transport layer and a Wi-Fi 802.11p as the physical layer. The visualization of the co-simulated vehicle (real node) in the 3D virtual scenario in the Webots simulation tool is also possible. An example of the execution of the co-simulation architecture can be seen in Figure 5.

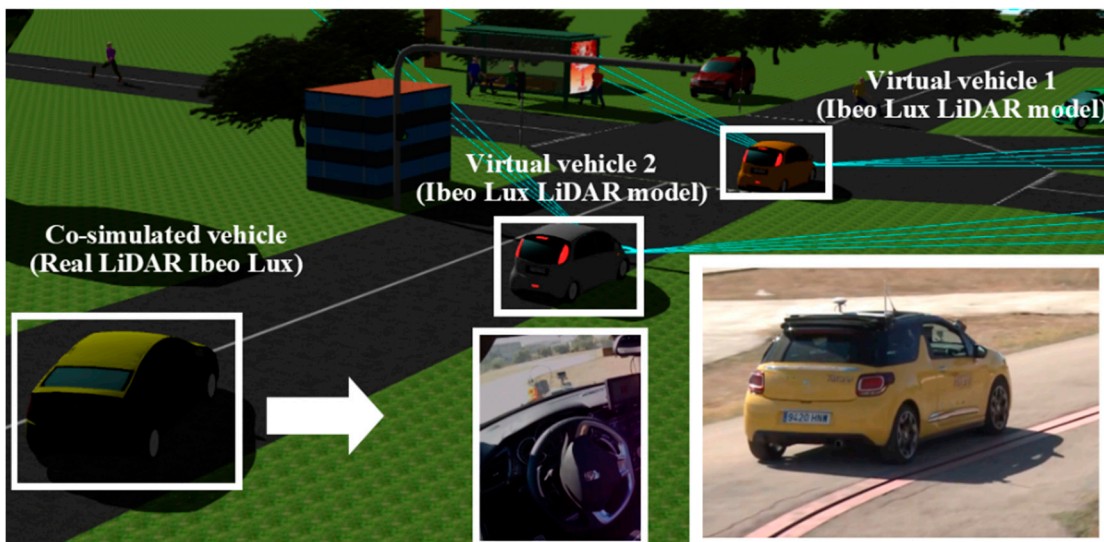

**Figure 5.** Data interchange between LiDAR sensors in (virtual and real) driving assistance scenarios in Webots for automobiles.

In addition, another application implemented in the IoT LiDAR-based collaborative map is the LiDAR self-testing methodology incorporated in each local computing node (autonomous vehicle), in order to evaluate the reliability of each IoT sensor in the network that will be described in Section 5.2.

### 5.1. Obstacle Detection in the IoT Application

The framework is implemented in an external application using Qt 5.10, which is an open-source widget toolkit for creating graphical user interfaces. The framework consists of an illustrated map updated in run time (see Figure 6a) and a database with the information on both virtual and real objects that are detected (see Figure 6b). The information contains the position, object type, and size of the obstacles in order to guarantee security and safety of the object detection process with a single sensor.

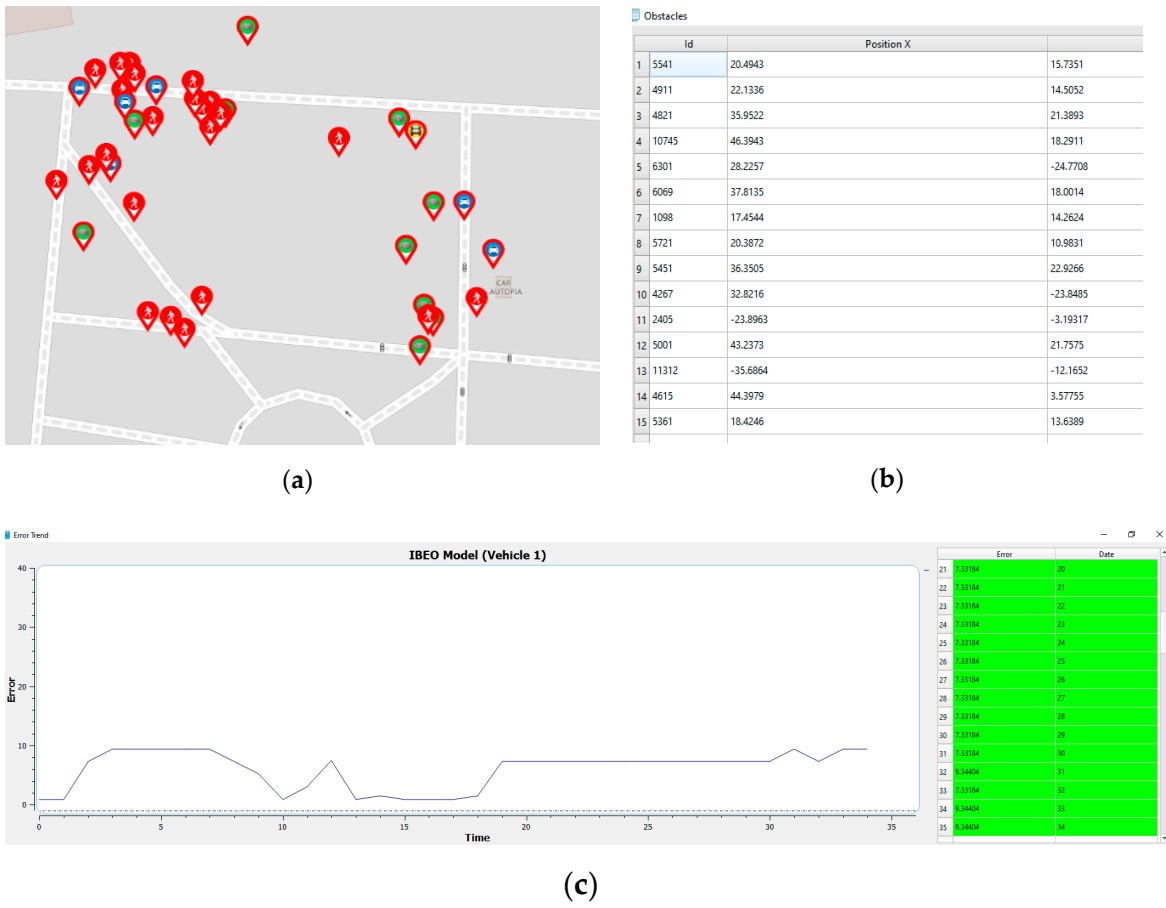

(**a**)　　　　　　　　　　　　　　　　　　(**b**)

(**c**)

**Figure 6.** (**a**) Collaborative mapping; (**b**) obstacles detected in the database; (**c**) LiDAR data for run-time accuracy error detection.

Figure 6 depicts the visual interface developed for the case study. Specifically, the collaborative map is globally updated in the main computing node. A partial area of this updated map can also be sent at the request of a local node. A set of computational procedures is in charge of adapting and transferring sensory information from Webots, virtual nodes with the Ibeo Lux sensor model, and the real node, real vehicles with the real Ibeo Lux sensor; and vice versa.

### 5.2. LiDAR Selft-testing Application

The external application also includes a LiDAR data self-testing methodology using the AI-based error-prediction models. Figure 6c shows the graphical interface that represents the estimated error with regard to time on the left-hand side. However, on the right-hand side, the admissible error threshold is observed, whereby, if it is exceeded, it must be requested that a decision be made regarding

the best performance of each model at any given time. Specifically, the results are focused on showing the improved performance of the IoT sensor network composed of each CPS element with each LiDAR sensor plus added IoT capabilities. To do so, a reliability prediction model dedicated to obtaining the accuracy error in obstacle detection is incorporated in each computing node.

### 5.2.1. Reliability Prediction Models

A reliability model is generated for both virtual and real IoT LiDAR sensors for predicting the accuracy error in obstacle detection. The procedure for developing these models is extracted from the methodology described in [27], with a different set of training data. In this study, a model-based procedure is used with a point-cloud clustering technique, in this case Density-Based Spatial Clustering of Applications with Noise (DBSCAN) [92]. In addition, an error-based prediction model library is implemented, highlighting AI-based model techniques, such as, Multilayer Perceptron Neural Network (MLP), k-Nearest Neighbors (k-NN), and Linear Regression (LR). A difference in the particular implementation described in this paper is that, while k-NN, MLP and LR were maintained, SVM is incorporate as a new technique to the modeling library [93–95].

### 5.2.2. Model Parametrization and Validation

With the aim of determining which model training strategy based on AI provides the best reliability prediction model, an experimental validation was performed. The training dataset was composed of 998 scenes for the model training and 250 scenes for the model validation. All of them were obtained from a simulation procedure. The data input consisted of geospatial statistics [70,96] that were extracted from the point cloud supplied by the LiDAR sensor, so that the models could generate the figures of merit in terms of accuracy error: DRMS and MRSE.

The four AI-based strategies considered can be summarized as follows. First, a multilayer perceptron neural network with backpropagation error (MLP) is selected with two hidden layers, each with five neurons and sigmoid activation functions, and an output layer with a lineal activation function, two neurons, and 5000 epochs. The initial value of the learning rate ($\mu$) is $10^{-3}$ with a decrease factor ratio of $10^{-1}$, an increase factor ratio of 10, and a maximum $\mu$ value of 1010. The minimum performance gradient was $10^{-7}$. The training process stop criteria is as follows: the maximum number of epochs (repetitions); goal performance minimization; the performance gradient below a minimum gradient; or, a $\mu$ value in excess of the maximum value. The second modeling technique is *k*-nearest neighbors (k-NN), with $k = 5$ and using Euclidean distance as the distance function. The third method is a lineal regression that was also obtained by minimizing the sum of squared differences between the predicted and the observed values. Finally, a support vector machine model is also obtained by means of data standardization and the radial basis function kernel.

### 5.2.3. Self-Learning-Based Decision-Making. Q-Learning Algorithm

The global or main computing node executes several parallel procedures in a specific self-learning module that uses a Q-learning algorithm (see Figure 7). On the one hand, a dataset for training by default is done in the global node. On the other hand, a knowledge database (warehouse) is also included, which can be updated in run time with the data provided by each local node. The self-learning strategy also runs in the global node in order to determine the best model behavior, when new traffic situations are generated by providing new point clouds (environment information).

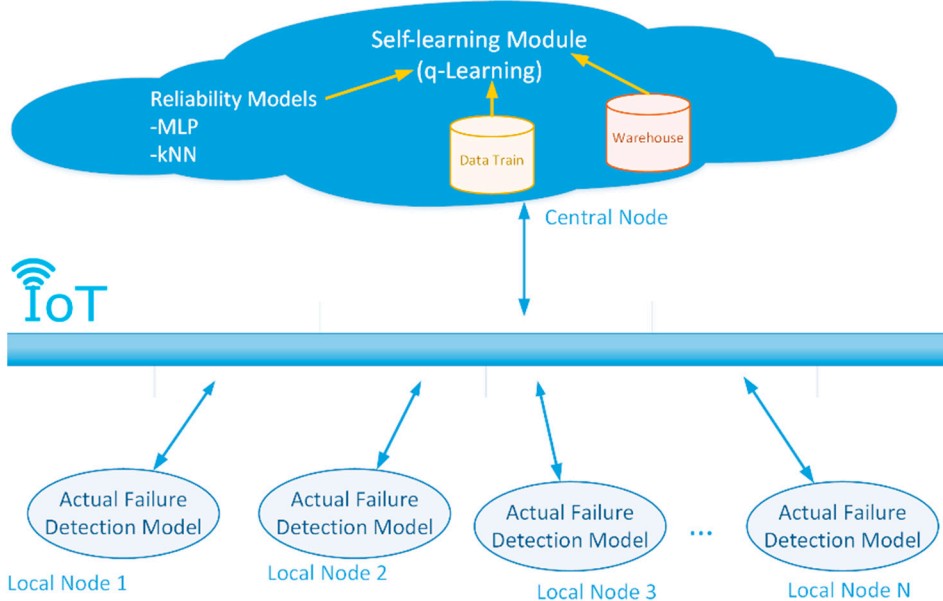

**Figure 7.** Flow diagram between the global node (self-learning module), IoT network, and local nodes (actual failure detection model).

The local node simulates the reliability model. When the error surpasses the 20% threshold in one the figures of merit (DRMS or MRSE), it is inferred that the current model is failing. A request is therefore made to the global module to establish whether there is a model that is working better. The decision for selecting the best prediction model available in the library is taken during the decision-making stage at each instant by means of a self-learning module, on the basis of the generalization capability and the accuracy of the model. The corresponding performance metrics are $R^2$ and RAE, respectively. In summary, based on this information flow and the previous prediction results (knowledge database), when a request from one of the local nodes is received and a new best model behavior is detected, the current error prediction model is then switched from MLP to k-NN, and vice versa.

## 6. Experimental Results

First, a comparative study was conducted on the basis of different AI techniques applied to model the prediction error obtained in objects detection. Spatial statistics, reported in [27], are the inputs for being aware of the condition of the point cloud in each scan provided by the LiDAR sensor. Subsequently, based on the results of this study, the best models are determined for the final evaluation. The final decision-making is done by means of a Q-learning strategy, which was selected for its good features among unsupervised learning methods [97]. This procedure predicts in advance which model has the best behavior on the basis of the available data information.

### 6.1. Reliability Model-based Validation

Table 2 shows the evaluation results obtained during the initial validation of each reliability model. Five error-based performance indices and two classification criteria were considered in the validation process: Mean Absolute Error (MAE); Root Mean Squared Error (RMSE); Relative Absolute Error (RAE); Root Relative Squared Error (RRSE); and, the coefficient of determination ($R^2$). Only the models generated with k-NN and MLP returned $R^2$ results higher than 90%. Key performance indices based on plane (DMRS) and space (MRSE) figures of merit.

**Table 2.** Key performance indices based on plane (DMRS) and space (MRSE) figures of merit.

| Model | MAE | | RMSE | | RAE | | RRSE | | $R^2$ | |
|---|---|---|---|---|---|---|---|---|---|---|
| | DMRS | MRSE | DMRS | MRSE | DMRS | MRSE | DMRS | MRSE | DMRS | MRSE |
| MLP | 0.0046 | 0.0035 | 1.275 | 1.270 | 0.187 | 0.188 | 0.395 | 0.392 | 0.933 | 0.933 |
| k-NN | 0.002 | 0.0002 | 1.014 | 1.010 | 0.114 | 0.114 | 0.371 | 0.365 | 0.963 | 0.961 |
| LR | 0.6781 | 0.6530 | 2.305 | 2.285 | 0.701 | 0.695 | 0.782 | 0.788 | 0.434 | 0.435 |
| SVM | 0.4735 | 0.4740 | 2.072 | 2.065 | 0.442 | 0.447 | 0.773 | 0.773 | 0.692 | 0.684 |

Figure 8 illustrates the behavior of the LiDAR error on the plane (DRMS) and space (MRSE) for each model (MLP, LN, k-NN and SVM) with regard to the validation data. The modeling techniques that provide the best performance are MLP and k-NN, according to the comparative study of the four modelling strategies, improving about 30% the performance indices with regard to LR and SVM. These two type models are then chosen for validating the decision-making module.

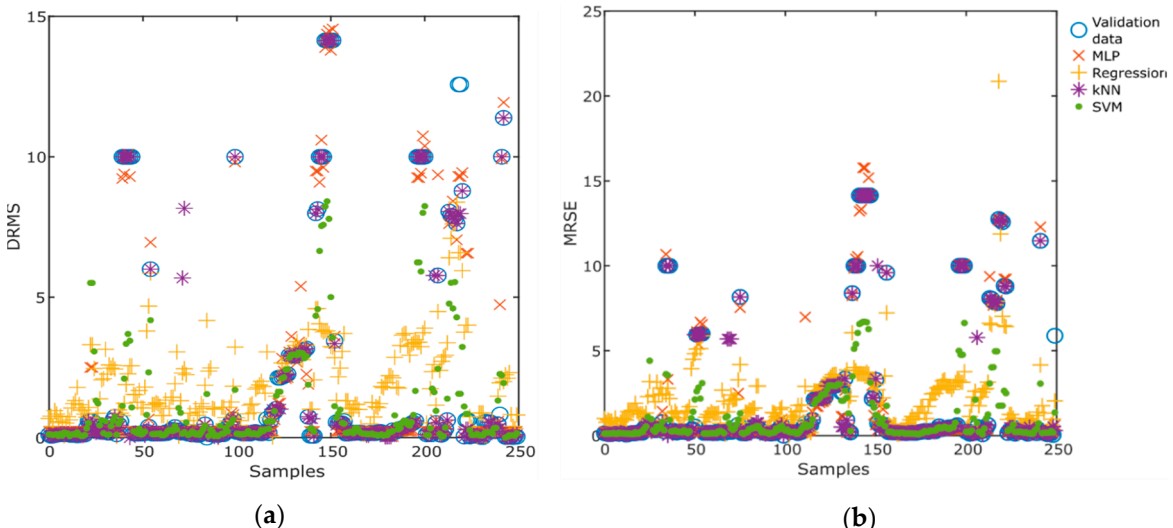

(**a**)            (**b**)

**Figure 8.** Behavior of LiDAR error of each model with regard to the validation data, (**a**) on the plane (DRMS) and (**b**) space (MRSE).

### 6.2. Self-Learning for Decision-Making Evaluation

Based on the previous knowledge generated by the rewards obtained from similar situations studied in the simulations, the ranges of the values of each reward are defined as a function of the prediction error of the reliability model, calculated during the simulation. The reward function is chosen for setting the best possible Q-value in 100 different scenes. Therefore, the function for updating the Q-values is [98]:

$$Q(s_{t+1}, a_{t+1}) \leftarrow Q_t(s_t, a_t) + \alpha_t \left( R_{t+1} + \gamma \max_{a \epsilon A} Q(S_{t+1}, a_t) - Q(S_t, a_t) \right) \tag{2}$$

where $s_t$ is the state in time $t$; $a_t$ is the action taken in time $t$; $R_{(t+1)}$ is the reward received after performing action $a_t$; $\alpha_t$ is the learning rate ($0 \leq \alpha_t \leq 1$); and $\gamma$ is the discount factor which trades off the importance of sooner-versus-later rewards. Table 3 lists the error reward matrix based on knowledge of the behavior of those prediction models.

**Table 3.** Q-learning reward matrix for error admissible threshold.

| $R^2$ | RAE | | | | |
|---|---|---|---|---|---|
| | 0–10% | 10–20% | 20–40% | 40–70% | >70% |
| 90–100% | 1 | 0.9 | 0.8 | 0.5 | 0.2 |
| 80–90% | 0.85 | 0.8 | 0.65 | 0.4 | 0.15 |
| 70–80% | 0.7 | 0.6 | 0.5 | 0.3 | 0.1 |
| 30–60% | 0.5 | 0.4 | 0.3 | 0.2 | 0.05 |
| 0–30% | 0.3 | 0.2 | 0.15 | 0.1 | 0.01 |

The decision-making is based on two of the main performance indices. First, the coefficient of determination ($R^2$) is taken into account, because it provides a measure of the generalization capacity of the model. The Relative Absolute Error (RAE), which is a measure of model accuracy, is also considered.

Finally, another simulation is performed in order to determine the quality of the Q-learning method for automatically selecting the best prediction model. One hundred new scenes captured by the LiDAR sensor were analyzed. The Q-learning method runs online in the main node. When new data becomes available, models contained in the library are evaluated. Once each model has been assessed, the performance indices are calculated, and the Q-learning determines the best reliability model between MLP and k-NN. Figure 9a,b depicts the behavior of RAE and $R^2$, respectively, for both reliability models.

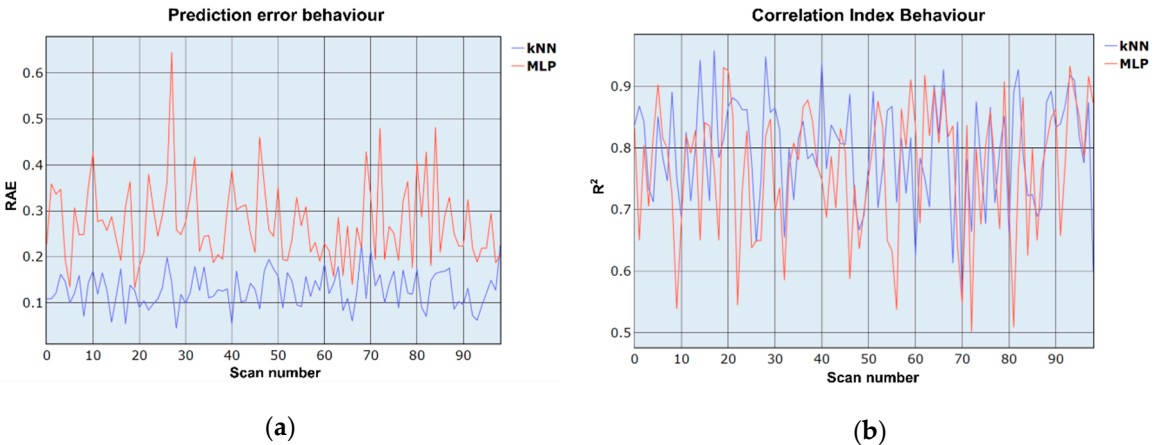

(**a**)                                    (**b**)

**Figure 9.** Behavior representation of both reliability models. (**a**) Prediction error behavior, RAE, and (**b**) correlation index behavior, $R^2$.

Figure 10 shows the Q-learning classification error matrix. In all considered cases, the best model chosen was k-NN 76 times and *MLP* 24 times. Analyzing the results obtained, the best model selected had a RAE between 0 and 20% and a $R^2$ above 80% in 61% of the scenarios. The system was able to guarantee models with a greater capability of generalization in 71% of the scenarios, based on a coefficient of determination higher than 80%. Overall, reliability can be predicted with an RAE of less than 40% and an $R^2$ of about 70% in 90% of the scenarios, which demonstrates the quality of the learning process. The models presented a low generalization, with a coefficient of determination of less than 70% in only 9% of the scenarios and an RAE greater than 40% in only 1%. Therefore, the Q-learning method that evaluates reliability on the basis of the prediction error model at each instant worked appropriately for calculating the best model that represented the LiDAR performance to a high degree of accuracy and that guaranteed the required levels of safety and reliability for automotive applications.

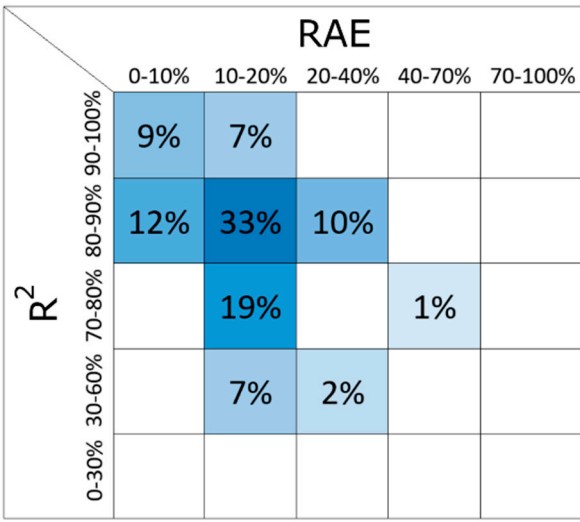

**Figure 10.** Q-learning classification error matrix.

## 7. Conclusions

This paper presents a review of the main research and innovation actions dealing with sensor reliability in cyber-physical systems supported by the European Commission in recent years. The review demonstrates that cyber-physical systems pose greater scientific and technical challenges due to the need for conjunctive approaches for evaluating sensor reliability using Internet-of-Things data. New research trends are focused on machine learning methods for improving metrics and sensor reliability in cyber-physical systems. However, the insufficient performance of big data conditioning and pre-filtering methods, the high computational cost and the ad-hoc parametrization of machine learning methods still limit generalization, real-time application and certification of solutions based on Internet-of-Things data for increasing sensor reliability.

This paper also presents a case study of sensor reliability assessment in autonomous driving scenario. The procedure for evaluating the reliability of IoT sensors in a cyber-physical system is proposed. A co-simulation framework is designed in order to enable real-time interaction between virtual and real sensors by running simulations in appropriate safety conditions. The case study consists of a IoT LiDAR-based collaborative map in order to validate the CPS-based co-simulation framework.

The assessment of the proposed method is divided into two parallel procedures. First, at a local level, each reliability model evaluates the condition of the IoT LiDAR sensor. The evaluation is carried out on the basis of AI-based models under 250 scenes. Based on the obtained results, only the multilayer perceptron and k-nearest neighbor methods were chosen for the validation of the decision-making module. Secondly, at a global level, a self-learning strategy for decision-making calculates the most appropriate behavior in the reliability model library, also in run time. Based on the results with 100 different scenarios, the Q-learning method improves the reliability of LiDAR sensor data with regard to using a single model at a local level.

Therefore, the proposed co-simulation framework serves to assess the performance of IoT LiDAR sensor data very accurately, guaranteeing safety and reliability in this autonomous driving scenario. These promising results serve as the basis for future work in validating the proposed method in real autonomous driving conditions.

**Author Contributions:** R.E.H., J.K. and S.S. reviewed all technical and scientific aspects of the article. A.V. and F.C. was in charge of the implementation of software application, the library models and the reinforcement learning algorithm. F.C. and A.V. designed and implemented the scenario, the external application and the LiDAR self-testing procedure, and drafted the paper.

**Funding:** This work was partially supported by the project Power2Power: Providing next-generation silicon-based power solutions in transport and machinery for significant decarbonisation in the next decade, funded by the Electronic Component Systems for European Leadership (ECSEL-JU) Joint Undertaking and the Ministry of

Science, Innovation and Universities (MICINN), under grant agreement No 826417. In addition, this work was also funded by the Spanish Ministry of Science, Innovation and Universities through the project COGDRIVE (DPI2017-86915-C3-1-R). Preparation of this publication was also partially co-financed by the Polish National Agency for Academic Exchange (NAWA) through the project: "Industry 4.0 in Production and Aeronautical Engineering (IPAE)".

**Acknowledgments:** The authors would like to thank the AUTOPIA group located at the Center for Automation and Robotics, jointly owned by the Spanish National Research Council and Technical University of Madrid.

**Conflicts of Interest:** The authors declare no conflict of interest.

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
