# Peer review of "Sensor Reliability in Cyber-Physical Systems Using Internet-of-Things Data: A Review and Case Study"

_remotesensing, doi:10.3390/rs11192252_

Round 1

Reviewer 1 Report

This paper introduces a method and an accompanying procedure for evaluating the 
reliability of IoT sensors in a CPS

LiDAR sensors in automotive application 
are not well motivated in the introduction.

It is not clear how IoT improves the reliability of LiDAR sensors.

The paper must consider a related work section in order to enable the novelty of the proposal.

Author Response

Firstly, authors wish to thank the reviewer for suggestions. The authors are very grateful for the constructive and detailed comments of the reviewer.

LiDAR sensors in automotive application are not well motivated in the introduction.

Following the comments of the reviewer, a paragraph that motivates the use of LiDAR sensor is added:

“Currently, two of the sensors that are mostly applied are the 3D stereoscopic camera and the LiDAR [12, 13]. Dozens of new applications are recently reported [14-16]. However, the computational load and the processing time for data fusion of 3D stereoscopic camera and the LiDAR sensors in automotive applications are very high [17]. Therefore, a possible solution is to use only the point cloud provided by the LiDAR with a model for predicting the reliability of the sensor data and only activate the support sensors when a possible failure is anticipated [18].”

It is not clear how IoT improves the reliability of LiDAR sensors.

In the introduction, the following paragraph has been added for clarify and justified the improvement introduced by IoT in the reliability of LiDAR sensors:

Due to it is very difficult to certify the real topology and distance of objects at a lower level of uncertainty, in most cases due to dead zones, object transparency, light reflection, weather conditions, and sensor failures, among others [8], the obtaining data in-parallel from other connected LiDAR sensors to guarantee reliability is become strictly necessary.  Therefore, new IoT-based technologies open a wide range of methodologies to estimate and ensure reliability in automotive applications.

In addition, in the conclusions, the third paragraph emphasizes the increase in reliability due to having a network of IoT sensors.

The paper must consider a related work section in order to enable the novelty of the proposal.

According to the reviewer´s recommendation, the authors have added a section titled “Reliability of sensors in cyber-physicals systems. A brief state-of-the-art review”.

Reviewer 2 Report

Authors proposed a procedure for evaluating the reliability of Internet-of-Things (IoT) sensors and the use of a Cyber-Physical System (CPS) for the implementation of that evaluation procedure to gauge reliability. The paper is well-written, however, there are some comments as follows:

Authors should add a section that compares the proposed method to the related works. The evaluation of the proposed method is not enough. The proposed method should be evaluated more clearly using proper methods. Conclusion should be more concise.

Author Response

The authors are very grateful for the reviewer's comment.

Authors should add a section that compares the proposed method to the related works.

According to the reviewer´s recommendation, the authors have added a section titled “Reliability of sensors in cyber-physicals systems. A brief state-of-the-art review”.

The evaluation of the proposed method is not enough. The proposed method should be evaluated more clearly using proper methods.

Following the reviewer's recommendation, the authors have improved the presentation of the results of the learning method (section 6.2. Self-learning for decision-making evaluation) in order to better explain the evaluation method.

Conclusion should be more concise.

Following the reviewer´s recommendations, the authors have modified the conclusions.

Reviewer 3 Report

The paper is very clear and describe very well the principal ideas behind the reliability of IoT sensors. The cosimulation used sound interesting and the use of Q learning  method result very innovative in this domain. 

Author Response

The authors are very grateful for the constructive and detailed comments of the reviewer.

Reviewer 4 Report

This is a well-structured paper, reporting novel and timely research results relevant to evaluating the reliability of sensors through utilization of IoT LiDAR data in a CPS.  The reliability assessment procedure described in this paper can find many applications, particularly, in the growing area of autonomous vehicles.

The paper is very well-written. It may be marginally improved by considering the following points.

While English language usage is very good, another round of proofreading can help. For example, a word like 'technique' seems to be missing in line 126. In line 148, the word 'assume' perhaps needs to be changed. Check Eq 2. The evaluation-related parts can be clearer on the use of AI, statistical methods, and Q-Learning.  More extensive comparative analysis of the reported methodology with existing approaches aiming for the same objectives can better show the novelty and the potential performance improvements.

Author Response

The authors are very grateful for the reviewer's comment.

a word like 'technique' seems to be missing in line 126.

The authors have included the word “technique” in Line 211, old line 126.

In line 148, the word 'assume' perhaps needs to be changed.

The authors have changed the word “assume” by “are increasing the relevance”

Check Eq 2.

The authors are checked and modified the equation thanks to the reviewer´s comment.

The evaluation-related parts can be clearer on the use of AI, statistical methods, and Q-Learning.

The authors are improved the structure and description of section 6 “Experimental results” in order to clarify the evaluation procedure.  After section title “6. Experimental results” the following paragraph explaining the evaluation procedure has been added.

“First, a comparative study was conducted on the basis of different AI techniques applied to model the prediction error obtained in objects detection. Spatial statisticians, reported in [98], are the inputs to be aware about the condition of the point cloud in each scan provided by the LiDAR sensor. Subsequently, from the result of this study, the best models are determined for the final evaluation. The final decision-making is done by means of a Q –learning strategy, selected for the sake of good features among unsupervised learning methods [99]. This procedure predicts in advance which model has a better behaviour according to the available data information.”

More extensive comparative analysis of the reported methodology with existing approaches aiming for the same objectives can better show the novelty and the potential performance improvements.

According to the reviewer´s recommendation, the authors have added a section titled “Reliability of sensors in cyber-physicals systems. A brief state-of-the-art review”. More than 100 papers have been considered in the review of the state of the art including 10 European projects in this field. Also, the experimental results have been modified in order to clarify the results. 

Round 2

Reviewer 2 Report

Authors applied all comments completely.